# A dream EEG and mentation database

Magneto/electroencephalography (M/EEG) studies of dreaming are an essential paradigm in the investigation of neurocognitive processes of human consciousness during sleep, but they are limited by the number of observations that can be collected per study. Dream research also involves substantial methodological and conceptual variability, which poses problems for the integration of results. To address these issues, here we present the DREAM database—an expanding collection of standardized datasets on human sleep M/EEG combined with dream report data—with an initial release comprising 20 datasets, 505 participants, and 2643 awakenings. Each awakening consists, at minimum, of sleep M/EEG ( ≥ 20 s, ≥100 Hz, ≥2 electrodes) up to the time of waking and a standardized dream report classification of the subject's experience during sleep. We observed that reports of conscious experiences can be predicted with objective features extracted from EEG recordings in both Rapid Eye Movement (REM) and non-REM (NREM) sleep. We also provide several examples of analyses, showcasing the database's high potential in paving the way for new research questions at a scale beyond the capacity of any single research group.

A third of the life of a healthy adult is spent asleep and, for a portion of that, dreaming. Throughout the night, during any stage of sleep, subjective conscious experiences can occur repeatedly—what we broadly call dreams. Dream studies, producing combined sleep and dream data, support various ongoing topics of research, including clinical (e.g., parasomnias), neurocognitive (e.g., learning and memory), and basic ones (e.g., the neural correlates of consciousness in dreaming). In this work, we present the Dream EEG and Mentation (DREAM) database, which gathers and standardizes multicentric recordings of electroencephalography (EEG), magnetoencephalography (MEG), and dream reports to advance the aforementioned research topics.

Commonly, people think about dreams as being immersed in a lifelike world, disconnected from the external environment, in which perceptually rich and narrative experiences may be had[1,2]. However, sleep is accompanied by many types of conscious experiences that do not match this notion. In the transition from wake to sleep and vice versa, bizarre perceptual experiences may be reported, which are known as hypnagogic and hypnopompic hallucinations, respectively[3–6]. Minimal experiences of static, fragmentary, unimodal percepts or simple thoughts are often called sleep mentation. Some dreams are lucid: when dreamers are aware that they are dreaming and

may be able to control the oneiric content[7]. Some other dreams include "false awakenings", during which participants believe they have awoken but are actually still dreaming, sometimes giving the sensation of a dream within a dream[8]. A dream can be called a nightmare when it includes experiencing disturbing emotions, such as intense fear, anxiety, or grief, which may trigger an awakening. Frequently occurring nightmares can be a sign of an underlying mental or neurological disorder, such as post-traumatic stress disorder or narcolepsy[9–12]. In addition, the so-called "white dreams" are reported by awakened participants as a dream experience that they are convinced occurred but cannot recall the specific contents of[13]. Finally, there are periods of sleep in which experience is reported to have been absent altogether. These reports either signify unconsciousness during sleep or lack of any recall of dream experiences. Notably, the above delineations are not always unambiguously recorded in sleep studies, making data reuse between different research topics difficult. For this paper, we will broadly define dreams as any experiences recalled from sleep.

Subjective verbal reports provided upon awakening are considered the gold standard for collecting data from dream experiences in current research practices. Yet, the information we can extract from verbal reports has limits and tradeoffs. Sleep must be interrupted in

✉e-mail: katja.valli@his.se; naotsugu.tsuchiya@monash.edu

order for participants to report; the quality of reports may be impaired due to confusion or sleepiness (e.g., sleep inertia), and the comprehensiveness of reports diminishes with time due to forgetting and the complex act of reporting[14,15]. As such, depending on the research question, many awakenings must be performed to produce reliable data. In the special and rare case of lucid dreaming, participants can produce rudimentary but objective signals without waking up, in the form of voluntary eye[16], facial, or respiration[17] movements, or fist clenching[18–20].

The classic stages of sleep were identified based on visual examination of the polysomnography (PSG), which includes the simultaneous recording of the electroencephalogram (EEG, for brain activity), electrooculogram (EOG, for eye movement), and electromyogram (EMG, for muscle activity)[21–23]. According to the American Academy of Sleep Medicine[24], wakefulness features low-amplitude, high-frequency EEG activity, active EOG, and high EMG tone. When individuals keep their eyelids closed and relax, alpha rhythms (8–12 Hz) predominate. NREM stage 1 (N1 or sleep onset) presents a slowing of the EEG toward theta waves (4–7 Hz), slow rolling eye movements in the EOG, and reduced EMG activity. NREM stage 2 (N2 or light sleep) is marked by sleep spindles (a transient 12–16 Hz oscillation) and K-complexes in the EEG, no eye movements in the EOG, and further decreased EMG tone. NREM stage 3 (N3, deep sleep, or slow-wave sleep), shows trains of high-amplitude delta waves (0.5–4 Hz) in the EEG, absent EOG activity, and minimal EMG tone. REM sleep is identified by low-amplitude, mid-low to high frequencies EEG activity, rapid eye movements in the EOG, muscle atonia in the EMG, and has been traditionally linked to dreaming.

Early sleep studies utilizing PSG suggested that dreams occur mostly during a distinct stage of sleep, the REM sleep, and only rarely during non-REM (NREM) sleep[25,26]. However, many subsequent studies have revealed the connection between dreaming and sleep stages to be more complex (e.g.[27],). It is now established that dream reports accompany about 85% of awakenings from REM sleep and about 40–60% from NREM sleep[28–30], varying as the night progresses[31,32]. While dream reports from NREM sleep tend to be less frequent and have less vivid and detailed content compared to those obtained from REM sleep, NREM dreams occurring at sleep onset (N1) can resemble REM dreams in vividness and bizarreness (despite having a shorter duration)[3,33], and morning dreams from both REM and NREM sleep may be indistinguishable from each other[34]. Generally, the frequency of dream recall in both NREM and REM sleep increases in the latter half of the night, and so does dream complexity[35,36]. Thus, REM sleep is neither necessary nor sufficient for dream experiences[37]; dreams can occur in neurophysiologically distinct stages of sleep.

Objective and consistent markers of dreaming, or dream recall, are still being investigated. Recent studies suggest that the occurrence of reported dreams has a spectral EEG marker consistent between and within sleep stages: a reduction of low frequency delta power[38–46]. Other frequency bands have been reported to be associated with certain aspects of dreams[47–55], and so have other features of EEG such as spindles or network properties[56–59]. In addition, different aspects of neural correlates of dreaming may be revealed by using MEG in addition to EEG because of their methodological differences. EEG studies investigating the neural correlates of specific contents of dreaming[42,60–64], revealed a homogeneity of the neural basis of subjective experiences between wakefulness and sleep. Similarly to the waking brain, affective processing in dreams is associated with frontal alpha asymmetry[63], dreams involving kinesthesia and movement are linked to inter-hemispheric coherence of beta oscillations[64], perceptual contents of dreams correlate with gamma activity in the occipital, temporal, and parietal regions, and thought-like mentation is associated with frontal activation in the gamma frequency range[42]. However, more research is needed to determine exact associations

between multidimensional EEG patterns, type of conscious experience and specific dream contents.

Science has not yet determined whether or not awareness is present in cases where behavioral reports are unavailable. This problem is most prevalent in everyday sleep, but the implications run deeper when considering other conditions involving unresponsiveness. We can have no ease of certainty about the absence of consciousness in patients diagnosed with unresponsive wakefulness syndrome[65], or the depth of general anesthesia needed to prevent individual intraoperative awareness[66]. Indeed, phenomenologically, such intraoperative awareness is often reported/described as intraoperative dreaming[67]. Given these phenomenological similarities, it is hypothesized that discovering the neural correlates of consciousness in dreaming would help solve these problems in detecting consciousness in the absence of reports. Combined sleep and dream data provide a promising paradigm for such research.

Unfortunately, sleep studies that include dream sampling protocols require a huge amount of time, skills, and resources. Only a few laboratories can afford to perform this kind of investigation. Moreover, such studies often have to rely on relatively small sample sizes. Indeed, previous neural correlates studies of dreaming have usually been performed on sample sizes in the order of tens of awakenings, thus leading to reservations about the robustness and generalization of findings[68] and slowing the pace of progress. Studies collecting hundreds of awakenings are rare (e.g., Refs. [42,44]). As such, there is a significant communal benefit to having a large database of existing dream and EEG datasets contributed by the research community available. Such collaborative efforts are multiplying in the field of neuroscience (e.g., Allen Institute for Brain Science, Stanford Center for Reproducible Neuroscience, Refs. [69,70]), but until the current project, there has been no corresponding endeavor specifically for neurobiological dream research. Instead, existing related databases offer either sleep EEG data without dream reports (e.g., the National Sleep Research Resource[71]) or dream report data without neurophysiological recordings (e.g., DreamBank.net[72] and the Sleep and Dream Database https://sleepanddreamdatabase.org/).

A major challenge that must be overcome to build a reuse-friendly database is ensuring that all datasets, which originate from primarily unrelated studies, are comparable. We, therefore, propose a unified dream classification system that delineates the types of dreams relevant to consciousness research to provide unambiguous and comparable categorizations between studies. We also implement a consistent, high standard for data formatting and documentation. Pertinent metadata are extracted from each datum in the collection and entered into the database for search and browsing.

The present work describes the Dream EEG and Mentation (DREAM) database, created to facilitate and stimulate dream research and collaborative studies using a neuroscientific approach. The DREAM database is released with an initial 20 data sets, composed of 505 participants and 2643 awakenings. Each datum represents PSG data of a sleep period leading up to awakening and a subjective report of the sleep experience. The database is openly accessible from https://monash.edu/dream-database. Furthermore, the database is open to new data contributions and will continue to expand indefinitely. It is maintained by a core team of scientists from different institutions on a voluntary basis.

## Results
In the following section, we demonstrate the use of our database by analyzing behavioral, experiential, and electrophysiological data. For a detailed description of the methods see Supplementary Methods 2. For this analysis, we checked critical metadata and selected relevant datasets using the database without performing a time-consuming download of huge amounts of data, demonstrating the usefulness of the dataset. From the DREAM database, we retrieved relevant data, such as

the dream report classification and sleep stage of the pre-report epoch. We also retrieved the age and sex of the participant if provided.

## Behavioral analyses

Using combined dream report classifications and manually scored sleep stages of the final epoch, provided by contributors as aggregated in the DREAM database Data records table (version 1), we performed statistical analyses for the relationship between sleep stage and dream experience. Based on the previous literature, we expected the relative frequency of sleep experience categories to be different between awakenings from different sleep stages (e.g., Nielsen, 2000[28]). Qualitatively, we expected deeper NREM sleep to be associated with less frequent reports of dreaming and REM sleep to be associated with more frequent reports of dreaming than NREM sleep.

We observed a dependence between the *Last sleep stage* and *Experience* (Table 4) with high statistical significance (chi-squared test, $X^2 = 120.9$, $df = 6$, $p < 10^{-15}$). We also observed the expected qualitative trend of deeper NREM sleep stages associated with fewer reports of "Experience", and an association between NREM sleep depth with reports of "No experience" and reports of "Experience without recall". The REM sleep stage was associated with fewer "No experience" and "Experience without recall" reports and more "Experience" reports than expected with the independence assumption (Supplementary Table 4).

The behavioral data were further analyzed using generalized linear mixed-effects modeling to account for the grouping effects of the experimental study and individual participants. See Supplementary Methods 2 for method's details and Supplementary Table 5 for the estimated fixed effects coefficients, which are visualized in Fig. 1.

Based on the fitted model's posterior distribution, the following inferences could be made with high confidence (>99% density in the region of interest). We found the expected qualitative trend of deeper NREM sleep stages associated with lower odds of "Experience" against "No experience" reports. There was no clear trend for the odds of "Experiences without recall" against "No experience" reports with respect to NREM depth. The odds of "Experience" against "No experience" reports for the REM sleep stage were higher than for N2 and lower than for N1.

## EEG analyses

We analyzed EEG recordings of selected DREAM datasets in two respects: power spectral density and automated sleep scoring. Of the datasets available at the time of the DREAM database's first version, we selected six datasets suitable for our EEG analyses: Oudiette N1Data, Zhang & Wamsley 2019, De Gennaro Multiple Awakenings, De Gennaro Young Adults, Tononi Serial Awakenings, and REM Turku (see Table 3 for details). For brevity, in our results, they are named following the corresponding article's first author: Lacaux, Zhang, Scarpelli 1, Scarpelli 2, Siclari, and Sikka, respectively. The rest of the datasets were excluded because of the use of nonstandard equipment (e.g., dry EEG headsets), the presence of stimuli during sleep, or the participants' demographics (e.g., younger or older participants). We focused on the 30 s of sleep before each dream report and used the *Records.csv* tables from each dataset to assign those epochs to their respective sleep stages and dream experiences. The three primary categories of dream experience were included—i.e., "Experience", "Experience without recall" and "No experience"—comprising a total of 1462 30-s epochs prior to a report. We performed minimal preprocessing where appropriate.

## Power spectral densities

We computed the power spectral densities (PSDs) associated with Wake (W), NREM, and REM sleep stages from one central electrode (C3 or C4). We found the stereotypical alpha peak (8–12 Hz) in W, the

sigma (spindle) peak (12–15 Hz) in NREM, and an increase in the theta range (4–8 Hz) in REM (Fig. 2).

## Automatic sleep scoring

We applied a previously validated automatic sleep scoring algorithm[73] to the six datasets to estimate hypnodensities, which are probability distributions of the scored sleep stage for a given epoch. As the algorithm operates on 15-s epochs rather than 30, we averaged hypnodensities over the last two 15-s epochs prior to a dream report. For this analysis, we used two EEG channels (a central and occipital channel), two EOG channels (left and right), and one chin EMG channel.

Results are given in Fig. 3 as confusion matrices against human scoring. The sleep scoring algorithm yielded an average accuracy above .72, which was close, despite the lack of context, to the inter-human scoring accuracy following the American Academy for Sleep Medicine (AASM) guidelines[22].

## Combined dream reports and EEG analysis

**Prediction of experience from EEG activity.** New automated algorithms to score sleep EEG recordings offer a probabilistic, rather than discrete, approach to sleep staging[73,74]. This probabilistic approach can reveal subtler differences that could be masked by the classical categorical view of sleep stages. We thus examined whether the dream report category was reflected in the probability of each sleep stage (hypnodensities, Fig. 4). We performed a Bayesian ANOVA on the hypnodensities, modeling the main effects of automatically-scored sleep stage and dream report and a random grouping effect for each dataset. Note that we reduced three NREM categories (N1, N2, and N3) into a single factor of NREM for this analysis. Overall, hypnodensities tracked human sleep scoring (i.e., the stage with the highest probability in the hypnodensity was congruent with human-scored sleep stages) with a significant main effect of sleep stage on hypnodensities (Bayes factor, $\log(BF_{inclusion}) = +\infty$). Hypnodensities were also modulated by the dream report category, with both a main effect of report category (Bayes factor, $\log(BF_{inclusion}) = 8.582$) and interaction between categories and sleep stages (Bayes factor, $\log(BF_{inclusion}) = 10.367$). Post-hoc estimates indicate that NREM epochs associated with dream experience show an increased probability mass toward W (non-overlapping 95% credible intervals; Bayes factor, $\log(BF_{model}) = 96.7$).

## Combined dream reports and EEG analysis

Finally, we quantified the amount of information needed to successfully classify dream experience ("Experience" or "No Experience") from EEG activity both in NREM and REM sleep. For this analysis, we used only three electrodes (F4, C4, and O2) that were commonly used in the six aforementioned studies. We extracted traditional PSD features and non-traditional features obtained from the *catch22* analytical pipeline[75] (see Supplementary Methods 2 for details). Catch22 provides 22 minimally redundant features that capture complex and nonlinear characteristics of time series. Only NREM (N1 + N2 + N3) and REM 30 s epochs prior to a report were considered in further analysis. NREM consisted of 778 epochs with 252 NE epochs and 456 E epochs. REM consisted of 342 epochs with 51 NE epochs and 282 E epochs. We computed three sets of features for each electrode:

1. PSD: Normalized spectral power on 6 different frequency bands: delta (0.5–4 Hz), theta (4.1–8), alpha (8.1–11 Hz), sigma (11.1–15 Hz), beta (15.1–20 Hz) and gamma (20.1–35 Hz). In total, 18 features.
2. Catch22$_{bb}$: catch22 features computed for the broadband (bb) EEG signal (0.5–35 Hz). In total, 66 features.
3. Catch22$_{bf}$: catch22 features computed for the EEG signal band-filtered (bf) on the 6 aforementioned frequency bands. In total, 396 features.

We found that for both NREM and REM "Experience" could be significantly discriminated (Bonferroni corrected $p < 0.001$, Wilcoxon rank sum test vs null distribution) from "No experience" using both PSD and catch22 features (Fig. 5). For NREM, the highest performance was obtained for PSD features (average AUC = 0.586, 5-th percentile = 0.505, 95th percentile = 0.608), while for REM the highest performance was obtained for Catch22$_{bf}$ (average AUC = 0.700, 5-th percentile = 0.663, 95-th percentile = 0.731). These results demonstrate that the DREAM database can be exploited to effectively investigate the neural correlates of dreaming. Furthermore, our results suggest that a more complex and nonlinear analysis –beyond PSD– may provide information relevant to decoding dreaming experiences from EEG recordings.

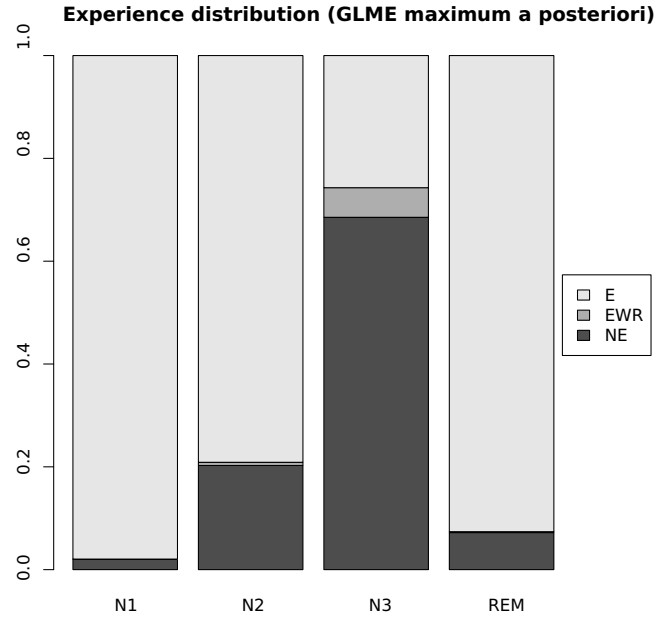

**Fig. 1 | Probability of reports per sleep stage.** *Y*-axis shows the maximum *a posteriori* probability for the three types of reports (E: "Experience", EWR: "Experience without recall", NE: "No experience"), split by each sleep stage (*x*-axis).

## Discussion

Many research areas share a need for, yet lack of, easily-accessible simultaneous sleep and dream data. One of our primary intentions for creating the DREAM database is to satisfy this need in the area of dream research. The DREAM database will: (1) enhance comparability across studies with an agreed dream report classification scheme (Tables 1 and 2), using combined sleep and dream data, (2) allow the data to be easily found and reused, and (3) help mitigate the problems related to small datasets and lack of statistical power. With the DREAM database, we present a variety of combined sleep and dream data sets in a novel, standardized, human- and machine-readable format, as recommended by the FAIR guiding principles[76]. Detailed metadata of all datasets are kept by the DREAM database proper and made openly accessible. Furthermore, we have been actively curating and expanding the database with new dataset submissions from community contributors and will continue to do so indefinitely.

The DREAM database is designed to enable researchers to more easily analyze data on automated end-to-end analysis pipelines as we demonstrated (Figs. 3–5). The database offers a great variety of recordings and includes dream reports collected both from REM and NREM sleep. Combined data unambiguously confirmed that dreaming prevalence decreases with sleep depth in NREM (Tables 3 and 4, Fig. 1). We also show that dreams in NREM sleep are associated with a higher probability of wake-like activity, suggesting that dreams could reflect a phenomenon of 'covert wake' akin to the covert REM hypothesis proposed previously[28] for NREM dreams (Figs. 3 and 4). Finally, we show that experience-relevant information can be decoded from EEG analysis both in REM and NREM sleep (Fig. 5).

The two latter findings should be further investigated in future studies, which goes beyond the scope of this paper. Our primary aim here is to present the database and demonstrate its use and potential. While useful on its own, the database and preliminary results will serve as the most promising starting point for future studies, to be published as the registered report or registered analysis format[77], which came about in response to the replicability crisis[78–81]. Pre-registered studies require the authors to precisely specify their methods in advance of their study's commencement, thus minimizing the risk of potential abuse of experimental and statistical methods—the so-called "data dredging" or "p-hacking". This database could inform sample sizes and narrow the hypotheses' space for new studies.

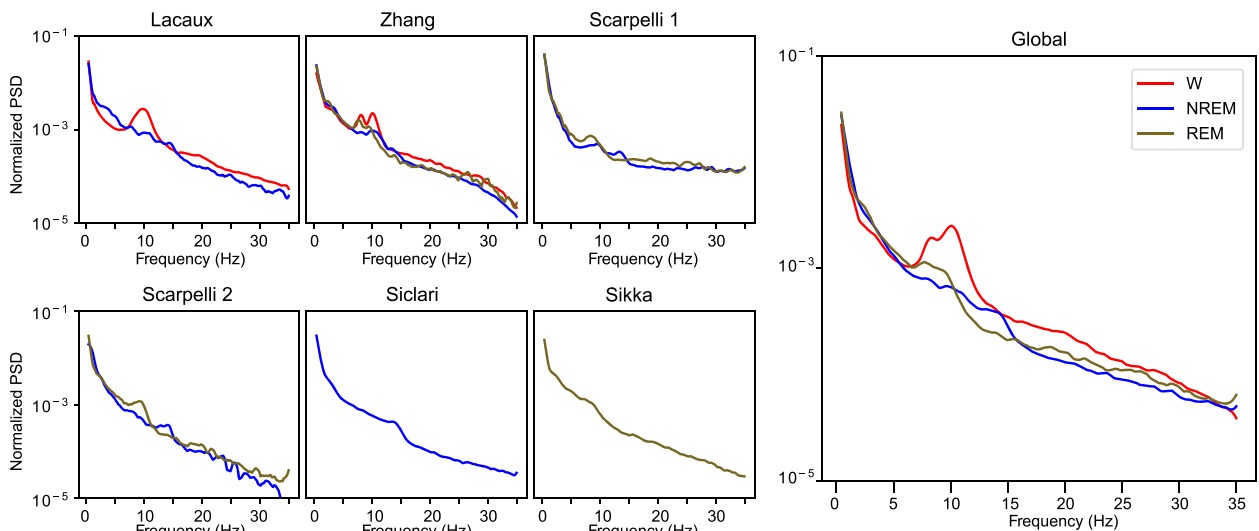

**Fig. 2 | Sleep stage power spectral densities.** Left (smaller) panels show the average normalized PSD for the 30-s epochs prior to a report for each database. Not all databases have all the sleep stages. The right panel shows the average of 1462 epochs from the 6 analyzed databases. Colors denote different sleep stages.

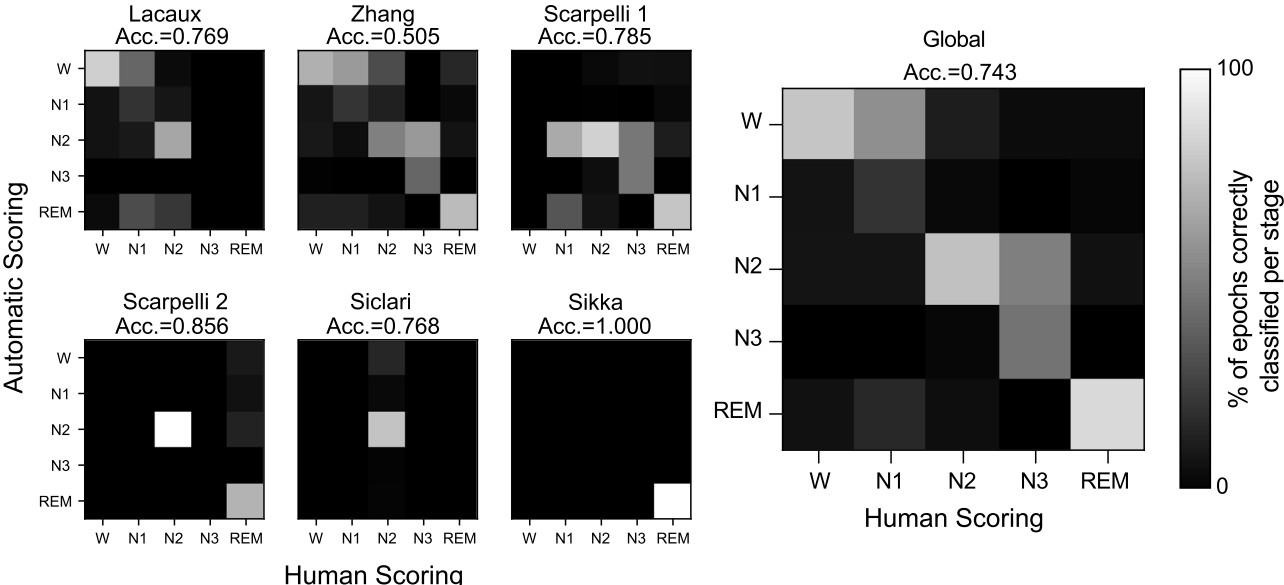

**Fig. 3 | Automatic sleep scoring.** The left panels show the confusion matrices for each database. Under each database name, the average subset accuracy score is shown (see Supplementary Methods 2 for details). The right panel shows the confusion matrix when all 1,462 epochs are considered, along with their respective average subset accuracy scores.

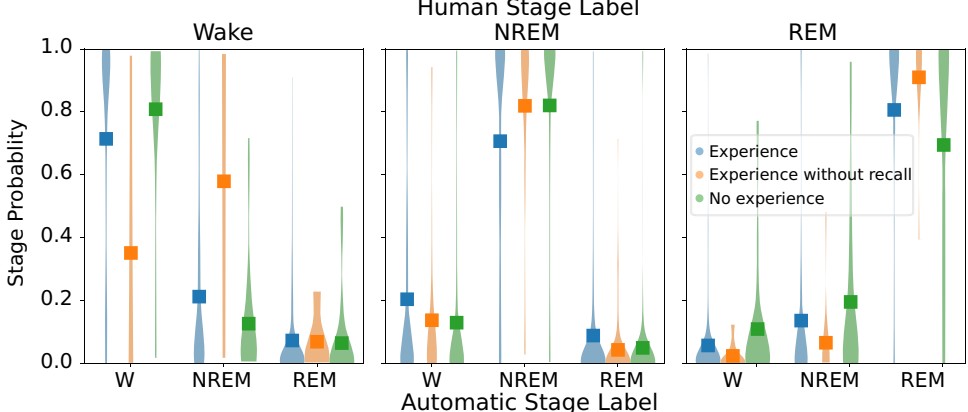

**Fig. 4 | Hypnodensities for each reduced human-scored sleep stage.** Distributions of hypnodensities per reduced sleep stage (where NREM = N1 + N2 + N3) for 1,462 pre-report epochs. All epochs belonging to the same sleep stage and dream experience were pooled together. Colors denote different categories of dream experience and squares the distribution average.

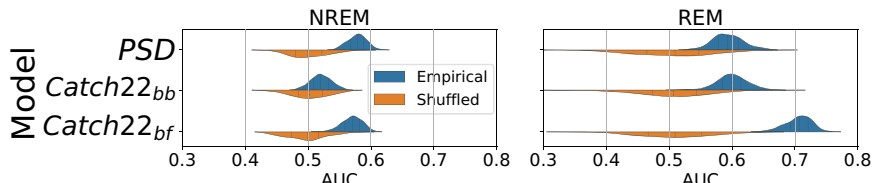

**Fig. 5 | Classification accuracy of dreaming experience (Experience vs. No experience) using EEG features.** Cross validation performance (in terms of area under the receiver operator characteristic curve, AUC, x-axis) from classifiers. Each of six classifiers was separately trained to discriminate dream experience between "Experience" and "No Experience" within NREM (left) and within REM (right) sleep, using a distinct set of features. Blue distributions (empirical) are the AUC values for test data from 200 random train-test splits of the data, while the orange ones (shuffled) are the AUC values for test data obtained by shuffling the labels for dreaming, serving as the null distribution. All empirical AUC distributions were significantly different from their respective null distributions (two-sided Bonferroni corrected Wilcoxon rank sum $p$-value < 0.001).

**Table 1 | Dream report classifications**

| Name | Definition |
|---|---|
| Experience | The participant reports having had experiences during sleep immediately prior to awakening and is able to recall some of their specific content. |
| Experience without recall | The participant reports having had experiences during sleep immediately prior to awakening, but has no recall of specific content, i.e., fails to recall any aspects of content (thoughts or imagery) while retaining a strong impression of having had experiences. This kind of mentation is also known as a "white dream." |
| No experience | The participant does not recall any experiences and has no impression that there would have been any experiences during sleep immediately prior to awakening. |

**Table 2 | Dream categorization rules and examples of their application**

| Dream categorization method | Example |
|---|---|
| If original classifications are analogous or fit into the standard definitions, then a direct mapping of the classifications is acceptable. | A study investigating dream recall categorized their dream reports as "Recall", "No recall" and "White dream". Here, these can be mapped to DREAM classifications "Experience", "No experience" and "Experience without recall". |
| If original classifications fit into none or more than one of the standard definitions—but the raw reports contain data that allow for reclassification—then a reclassification is provided in addition to the original classifications. | A study investigating dream recall categorized their dream reports as "Recall" and "No recall". As their raw dream report data were collected as free verbal reports, the "No recall" reports here can be reclassified to either "No experience", or "Experience without recall". |
| If original categories do not fit into the standard definitions unambiguously—and the raw reports do not contain data that allow for reclassification—then the original classification is provided, and its categories are interpreted as the combination of all standard definitions that could apply to it. | A study investigating dream recall categorized their dream reports as "Recall" and "No recall". Raw dream report data provided no additional information to allow recategorization of "No recall" reports to neither "No experience" nor "Experience without recall". Here, "Recall" can be mapped to the DREAM classification "Experience", and "No recall" can be mapped to the combined classification "Experience without recall or No experience". |

Participating journals are also required to make their decision to publish a registered study—based on the merit of the study's methods, before the study can commence and regardless of its eventual results—thus reducing the file drawer effect (i.e., not publishing null results that can potentially contradict previous findings). As combined sleep and dream data is relatively expensive to produce, increasing its sharing can also help boost the number of replications and novel analyses that can be performed, accelerating scientific progress.

Sleep research is attracting attention as it can host new experimental paradigms for investigating the neural correlates of consciousness, like the no-report within-state approach[82]. Dreams represent instances of human experience that largely have no direct sensory input from, nor observable behavioral output to, the external world. Thus, the observed neural processes for conscious experience are much less attributable to confounding factors such as preconscious sensory processing or task-related activity[82–87]. By contrasting dreamless, unconscious sleep with dreamful, conscious sleep, the distinction between levels of consciousness within the same brain state becomes possible.

As the database continues to grow, we hope it can help answer some of the remaining basic questions about sleep and dreams. Why are we sometimes conscious and sometimes unconscious during sleep? Why do some people remember their dreams every day while others have very low dream recall frequency? Why do so few individuals regularly have lucid dreams, while others can realize they are dreaming and control their oneiric content? Ultimately, discovering the neural correlates of specific contents of dreaming, which may require substantially more data than the classification of the presence or absence of dreaming experience, will play a significant role in understanding the neural mechanisms of consciousness. We look forward to seeing how future studies utilize the datasets presented here to tackle these and other questions and possibly identify new lines of research.

## Methods
### Ethics
The Dream EEG and Mentation database project was approved by the Monash University Human Research Ethics Committee (Project ID 31269). Furthermore, the authors of the originating studies of DREAM

datasets confirm that ethical approval for their respective studies were obtained from the relevant ethical committees, that participants provided informed consent and consented to sharing and secondary use of the data. The corresponding authors of these and future studies included by the database are obliged to protect participants' rights by one or a combination of: obtaining the participants' consent, ensuring the dataset is published with the appropriate accessibility, redacting data, and irreversibly de-identifying the data.

### Dataset selection criteria
The DREAM database has an ongoing call for data consisting of PSG of human participants prior to waking and associated reports of conscious experiences (or lack of such) of that sleep period. The selection criteria were arrived at by consensus among the researchers who answered our first call for participation.

The recommended minimal technical standards are:
11. At least two referenced EEG electrodes localizable by the 10-5 system;
12. At least 20 s of continuous sleep recorded up until the moment of awakening;
13. At least 100 Hz EEG sampling rate;
14. Raw data or minimally preprocessed data.

We also ask data contributors to include the following information.
21. Dream report classification for each datum (see Table 1) and the original dream report or data concerning it if possible;
22. Dream report categorization protocol, if nonstandard (see Table 2);
23. De-identified participant labels for each datum;
24. Treatment/condition/experimental group labels for each datum, if any;
25. Any notable observations or artifacts;
26. A description of the study protocols in enough detail to allow the replication of experimental procedures;
27. Special instructions for decoding data files, if any.

Dream reports may include oral report transcriptions, written reports by study participants, interview responses, and inventory

**Table 3 | The first set of datasets presented by the DREAM database**

| Dataset | Description | No. of awakenings | No. of participants |
|---|---|---|---|
| *Noreika Motor tDCS*, Noreika et al. | Bihemispheric tDCS over the sensorimotor cortex. REM-awakened free dream reports and bodily experiences questionnaire. | 49 | 10 |
| *Older adults*, De Gennaro & Scarpelli | Healthy, older adult participants. Spontaneous morning-awakened structured dream reports. | 40 | 40 |
| *Multiple awakenings*, De Gennaro & Scarpelli | Multiple awakenings within REM and NREM sleep. Awakened 2-alternative dream recall reports. | 489 | 20 |
| *Children Dreaming*, De Gennaro & Scarpelli | 9–14 year-old participants with developmental dyslexia or controls. Morning-awakened sleep and dream diary reports. | 30 | 30 |
| *Dream Young Adults*, De Gennaro & Scarpelli | Awakenings from NREM Stage 2 and REM sleep. Sleep and dream diary, and 2-alternative dream recall reports. | 65 | 65 |
| *Sleep Talking*, De Gennaro & Scarpelli | Healthy frequent sleep talkers. Morning awakenings following sleep talking. Self-reported sleep and dream logs. | 22 | 12 |
| *TWC USA*, Konkoly et al. | Participants trained to report lucid dreaming with eye movements, and to become lucid with external signal. Awakened structured interview dream reports. | 33 | 19 |
| *REM Turku*, Sikka et al. | Multiple REM awakenings. Verbal dream reports and emotional content questionnaire. | 134 | 18 |
| *LODE*, Elce et al. | Home sleep recordings with portable EEG and actigraphy devices. Spontaneous morning-awakened verbal dream reports. | 190 | 28 |
| *SCANDataset*, Eichenlaub, van Rijn & Blagrove | Slow waves and REM sleep awakenings. Awakened verbal dream reports. | 85 | 18 |
| *Oudiette N1Data*, Lacaux et al. | Daytime naps following creative task with multiple awakenings. Free mentation reports. | 252 | 63 |
| *Zhang & Wamsley 2019*, Zhang & Wamsley | Multiple sleep onset and mixed-stage awakenings following spatial learning task. Free verbal dream reports. | 308 | 28 |
| *Tononi Serial Awakenings*, Siclari et al. | Multiple NREM stage 2 awakenings. Awakened verbal dream reports. | 261 | 36 |
| *DATA1*, Noreika et al. | Multiple NREM Stage 2 and 3 awakenings. Awakened structured interview dream reports. | 324 | 10 |
| *Dream Database from Donders*, Demirel, Gott & Dresler | Dreaming, lucid dreaming and brain-computer interface studies. Awakened structured interview dream reports. | 7 | 6 |
| *Brain Institute - Federal University of Rio Grande do Norte*, Araujo et al. | Healthy participants. Morning-awakened free verbal dream reports. | 41 | 41 |
| *Aamodt evening sleep*, Aamodt et al. | Multiple, mostly NREM Stage 2 evening awakenings. Awakened structured interview dream reports. | 158 | 27 |
| *Aamodt morning sleep*, Aamodt et al. | Multiple, mostly NREM Stage 2 morning awakenings. Awakened structured interview dream reports. | 97 | 16 |
| *Kumral et al., 2023*, Kumral et al. | Participants enter sleep while listening to an audio book. Multiple awakenings. Awakened structured interview dream reports. | 66 | 19 |
| *MEG Kyushu*, Motomura, Takeichi et al. | Multiple NREM Stage 1 and 2 awakenings with simultaneous EEG and magnetoencephalography. Awakened semi-structured interview dream reports. | 31 | 1 |

**Table 4 | Sleep stage vs. Experience contingency table**

|  | No experience | Experience without recall | Experience | Total |
|---|---|---|---|---|
| **N1** | 12 (11%) | 1 (1%) | 97 (88%) | 110 (100%) |
| **N2** | 308 (36%) | 68 (8%) | 485 (56%) | 861 (100%) |
| **N3/NREM3/NREM4** | 26 (41%) | 7 (11%) | 31 (48%) | 64 (100%) |
| **REM** | 87 (17%) | 12 (2%) | 416 (81%) | 515 (100%) |
| **Total** | 433 (28%) | 88 (6%) | 1029 (66%) | 1550 (100%) |

Percentages are independent per row. See Supplementary Table 4 for results on deviance from the assumption of independence.

scores, but the minimum requirement is that the reports are categorized, where possible, into one of the following three classifications by the contributor: "experience", "experience without recall", and "no experience" or otherwise in combined classifications (where data are ambiguous). We describe the three main classifications in Table 1 (see below).

Data contributors are required to make their datasets available, either fully open-access or mediated-access, depending on what was approved by their own ethics committee.

### Dream report categorization

We have defined dreaming to have a very broad meaning here, both for simplicity and for inclusivity with regard to the database. Our definition of dreaming includes all forms of conscious experience occurring within any stage of sleep that the participant can recall, regardless of their specific contents. It does not distinguish between immersive, multisensory, emotional and narratively complex experiences and experiences involving only thinking, mentation, or isolated forms of imagery—for example, visual or auditory. Our database uses a unified

system for classifying dream reports into one, or a combination, of three ordinal categories: "experience", "experience without recall" (also known as "white dreams"), and "no experience." Their definitions are given in Table 1. As the contributed datasets can include additional, detailed subjective reports, it is possible to reclassify a subset of the datasets using a different set of definitions to the extent that the information is made available.

Since not all datasets may have originally collected and categorized data according to DREAM's definitions, the standardized dream report classification may have been recategorized from an original categorization or raw data. The procedures used for obtaining these are given in the dataset's description if applicable. Specifically, we applied the following rules (see also Table 2).

1. If original categories are analogous or fit into the standard definitions, then a direct mapping of the categories is acceptable.
2. If original categories fit into none or more than one of the standard definitions–but the raw reports contain data that allow for reclassification–then a reclassification is provided in addition to the original classification.
3. If original categories do not fit into the standard definitions unambiguously–and the raw reports do not contain data that allow for reclassification–then the original classification is provided and its categories are interpreted as the combination of all standard definitions that could apply to it.

### Data postprocessing
All datasets protect their subject's confidentiality by redacting or removing names, dates of birth, addresses or other personally identifiable information.

PSGs including annotations are formatted as European Data Format "plus" (EDF + ) files. No further processing is applied to the signal post digitization unless otherwise specified in the study description, and any relevant processing of PSGs during their collection are noted.

### Data Records
2643 data were collected in 20 separate studies. Descriptions of the individual datasets are given in Table 3. The registry of all datasets and their metadata constitute the main database and are searchable and downloadable at https://monash.edu/dream-database open access. Dataset packages are separately available either by open access or by request from the data contributor.

The DREAM database continues to be open to new contributions. Potential data contributors may visit the main website to receive instructions on how to format and upload their datasets.

### Dataset format
Each dataset is packaged in a standardized file directory structure convention. PSG data are found in the "Data/PSG" subdirectory, dream report data are found in the "Data/Reports" subdirectory, and other miscellaneous data are found under the "Data" directory. Metadata for each awakening (see Supplementary Table 1) and the study's experimental description are found in the root directory.

PSG data are presented in European Data Format (EDF)–compatible files, where each file contains data prior to a report. The recording length of the EDF file depends on the study design and is specified for each dataset. The Subject ID and Case ID values for each datum are encoded in the EDF header, specifically in the local patient identification field, whose format is analogous to the EDF+ standards. The Subject ID is given in the patient code subfield, and the Case ID is given in the patient name subfield.

### Database format
The database records metadata of each awakening from every study in the table "Data records" as a single record with 20 fields as listed in Supplementary Table 2. Additionally, metadata of each unique study of origin are recorded in the table "Datasets" as a single record with 31 fields listed in Supplementary Table 3.

These records can be downloaded as whole, are open access and searchable, allowing researchers to discover data relevant to their work easily. The database is updated when new datasets are added, or amendments are made.

### Technical validation
PSG artifacts were detected via visual inspection and noted by data contributors. From this, data that were found to contain =['<10 s of artifact-free EEG in the final 20 s were excluded from the datasets.

Database curators checked datasets via a combination of manual and automated checks for adherence to the DREAM dataset package standard. This included proper data quality (see Supplementary Methods 1), file directory structure, data file formatting, and metadata consistency.

Note also that manual sleep scoring can label a given epoch as Wake even before the participants were awoken (e.g., in serial awakening paradigms). In fact, there is abundant evidence for subjective-objective sleep discrepancies[88]. This happens in the context of our data, where either manual or automated EEG-based (objective) sleep stage classification labels an epoch as Wake, but during the same epoch, the participant reports to be asleep. For completeness, we have also included these subjective-objective discrepant epochs, as they may hold relevant EEG activity for dream experience. However, special attention should be paid to these epochs.

### Reporting summary
Further information on research design is available in the Nature Portfolio Reporting Summary linked to this article.

## Data availability
The DREAM database can be found from the landing page at https://monash.edu/dream-database or accessed directly at https://doi.org/10.26180/22133105. All DREAM datasets with open accessibility may be freely downloaded from the internet at their respective URL links recorded in the "Data URL" field of the "Datasets" table of the database (https://bridges.monash.edu/articles/dataset/The_DREAM_database/22133105?file=49774971). A detailed explanation of the columns of this table can be found on Supplementary Table S3. For datasets with more restricted access, a request for access may be made of the dataset's corresponding author; details as to the extent of the restrictions, whether there are individual sharing agreements and a time frame of response to requests are given in the respective "Data restriction note" field of the "Datasets" table. Corresponding authors are recorded as uniquely identifying keys in the respective "Corresponding contributor ID" fields; the keys can be looked up in the "Key ID" field of the "People" table of the database, where you can find their contact information. A description for each dataset in the DREAM database at the time of writing is given below, in Table 3.

## Code availability
Code to replicate the behavioral analyses of Fig. 1 and Table 4 can be found at https://doi.org/10.26180/29124026. Example code to run the analysis on Fig. 5 can be found at https://doi.org/10.5281/zenodo.15234845.

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

## Acknowledgements

WW and NT were supported by National Health Medical Research Council (APP1183280) and by Japan Society for the Promotion of Science, Grant-in-Aid for Transformative Research Areas (N.T. 23H04829, 23H04830). JW, TA, and NT were supported by the Australian Research Council (NT DP240102680). G.B, V.E, D.B, M.B and G.A. were supported by the BIAL Foundation Grant Number #091/2020.

## Author contributions

The DREAM database was conceived and organized by T.A., R.H., V.N., N.T., K.V., J.M.W., N.D and W.W. as the DREAM core team, which contributed equally. W.W. and R.H. drafted the paper. R.H., T.A. and W.W. performed the data analysis. K.C.A., D.B.D.A., I.A., S.A., G.A., B.B., M.B., D.B., G.B., M.B., Ç.D., M.D., J.B.E., V.E., S.G., L.D.G., J.G., C.H., B.E.J., K.R.K., D.K., C.L., J.J.L., B.L., R.M., S.A.M.R., Y.M., A.S.N., D.O., F.P.F., J.P., K.A.P., L.P., A.R., E.V.R., S.S., M.S., S.F.S., F.S., P.S., J.F.S., H.T., K.V., E.J.W., J.Zhang., J.Zhao contributed to the database. All of the contributors at the time of publication are listed as authors in alphabetical order except for the two co-first and the last authors. All of the authors approved the final version of the manuscript.

## Funding

## Competing interests
The authors declare no competing interests.

## Additional information

William Wong[1,39,40], Rubén Herzog[2,39,40], Kátia Cristine Andrade[3], Thomas Andrillon[2,4,40], Draulio Barros de Araujo[3], Isabelle Arnulf[2], Somayeh Ataei[5,6,7], Giulia Avvenuti[8], Benjamin Baird[9], Michele Bellesi[10,11], Damiana Bergamo[8,12], Giulio Bernardi[8], Mark Blagrove[13], Nicolas Decat[2,40], Çağatay Demirel[6], Martin Dresler[7], Jean-Baptiste Eichenlaub[14], Valentina Elce[8], Steffen Gais[15], Luigi De Gennaro[16], Jarrod Gott[7], Chihiro Hiramatsu[17], Bjørn Erik Juel[18,19], Karen R. Konkoly[20], Deniz Kumral[21], Célia Lacaux[2], Joshua J. LaRocque[22], Bigna Lenggenhager[23], Remington Mallett[20], Sérgio Arthuro Mota-Rolim[3], Yuki Motomura[17], Andre Sevenius Nilsen[18], Valdas Noreika[24,40], Delphine Oudiette[2], Fernanda Palhano-Fontes[3], Jessica Palmieri[21], Ken A. Paller[20], Lampros Perogamvros[25], Antti Revonsuo[26,27], Elaine van Rijn[28], Serena Scarpelli[6], Monika Schönauer[21], Sarah F. Schoch[6,7,29], Francesca Siclari[30,31], Pilleriin Sikka[26,27,32,33], Johan Frederik Storm[18], Hiroshige Takeichi[34], Katja Valli[26,27,40]✉, Erin J. Wamsley[35], Jennifer M. Windt[4,36,40], Jing Zhang[37], Jialin Zhao[6] & Naotsugu Tsuchiya[1,38,40]✉

[1]School of Psychological Sciences and Turner Institute for Brain and Mental Health, Monash University, Clayton, Australia. [2]Institut du Cerveau - Paris Brain Institute - ICM, Inserm, CNRS, APHP, Hôpital de la Pitié Salpêtrière, Sorbonne Université, Paris, France. [3]Brain Institute and Onofre Lopes University Hospital, Federal University of Rio Grande do Norte, Natal, Brazil. [4]Monash Centre for Consciousness & Contemplative Studies, Monash University, Melbourne, Australia. [5]Department of Neuropsychology, Faculty of Psychology, Ruhr University Bochum, Bochum, Germany. [6]Donders Institute for Brain, Cognition and Behaviour, Nijmegen, Netherlands. [7]Radboud University Medical Center, Donders Institute for Brain, Cognition and Behavior, Nijmegen, Netherlands. [8]MoMiLab Research Unit, IMT School for Advanced Studies Lucca, Lucca, Italy. [9]Department of Psychology, The University of Texas at Austin, Austin, USA. [10]School of Physiology, Pharmacology and Neuroscience, University of Bristol, Bristol, UK. [11]School of Biosciences and Veterinary Medicine, University of Camerino, Camerino, MC, Italy. [12]Department of General Psychology, University of Padova, Padova, Italy. [13]Sleep Laboratory, School of Psychology, Swansea University, Swansea, UK. [14]Univ. Grenoble Alpes, Univ. Savoie Mont Blanc, CNRS, LPNC, Grenoble, France; Institut Universitaire de France (IUF), Paris, France. [15]Institute of Medical Psychology and Behavioral Neurobiology, University of Tübingen, Tübingen, Germany. [16]Department of Psychology, University of Rome Sapienza, Rome, Italy. [17]Faculty of Design, Kyushu University, Fukuoka, Japan. [18]Section for Physiology, MolMed Dpt., Institute of Basal Medical Sciences, Faculty of Medicine, University of Oslo, Oslo, Norway. [19]Center for Sleep and Consciousness, University of Wisconsin–, Madison, WI, USA. [20]Department of Psychology and Cognitive Neuroscience Program, Northwestern University, Evanston, IL, USA. [21]Institute of Psychology, Neuropsychology, University of Freiburg, Freiburg, Germany. [22]Department of Neurology, Medical College of Wisconsin, Milwaukee, WI, USA. [23]Association for independent research, Zurich, Switzerland. [24]Centre for Brain and Behaviour, Department of Psychology, School of Biological and Behavioural Sciences, Queen Mary University of London, London, UK. [25]University of Geneva, Geneva, Switzerland. [26]Department of Psychology and Speech-Language Pathology, and Turku Brain and Mind Center, University of Turku, Turku, Finland. [27]Department of Cognitive Neuroscience and Philosophy, University of Skövde, Skövde, Sweden. [28]Department of Psychology, Swansea University, Swansea, UK. [29]Center of Competence Sleep & Health Zurich, University of Zurich, Zurich, Switzerland. [30]Netherlands Institute for Neuroscience, Amsterdam, Netherlands. [31]University of Lausanne, Lausanne, Switzerland. [32]Department of Anesthesiology, Perioperative and Pain Medicine, School of Medicine, Stanford University, Stanford, USA. [33]Department of Psychology, Stanford University, Stanford, USA. [34]Open Systems Information Science Team, Advanced Data Science Project, RIKEN Information R&D and Strategy Headquarters (R-IH), RIKEN, Yokohama, Japan. [35]Department of Psychology and Program in Neuroscience, Furman University, Greenville, SC, USA. [36]Department of Philosophy, Monash University, Clayton, Australia. [37]Department of Cognitive Sciences, University of California, Irvine, CA, USA. [38]Laboratory of Qualia Structure, ATR Computational Neuroscience Laboratories, Kyoto, Japan. [39]These authors contributed equally: William Wong, Rubén Herzog. [40]These authors jointly supervised this work: William Wong, Rubén Herzog, Thomas Andrillon, Nicolas Decat, Valdas Noreika, Katja Valli, Jennifer M. Windt, and Naotsugu Tsuchiya. ✉e-mail: katja.valli@his.se; naotsugu.tsuchiya@monash.edu

