## [Transparent Peer Review file · Nature Communications]

DREAM: A Dream EEG and Mentation database

Corresponding Author: Professor Naotsugu Tsuchiya

Version 0:

Reviewer comments:

Reviewer #1

(Remarks to the Author)

I confirm my positive opinion on the manuscript. The authors addressed my previous concerns.

Reviewer #2

(Remarks to the Author)

Apologies for the brief response:

1. I disagree this database will impact detection of anaesthetic awareness. I would remove all mention of anaesthesia. The pharmacology and interaction with surgical stimulation/artefact make it highly unlikely this database (of sleep studies) will help with that issue. That does not negate the utility of the database.
2. This is underscored by the very low classification accuracy of the catch-22 results. I think the findings in this paper are largely incremental. Apologies I had not understood the previous submission did not require analyses, however I do not think the attached analyses, based on a limited number of electrodes, represent a major shift in the science of the study of consciousness.
3. That said, I don't have much else to critique and I think it is a worthy effort to make these data publicly available.

Reviewer #3

(Remarks to the Author)

The authors have constructed a large database of dream reports and EEG/MEG recordings by combining data from 20 studies. In addition, they propose to grow the database by incorporating new studies and offer a suggested set of technical details. This is a very exciting project for, as the authors note, this kind of data is difficult to collect at scale. Dreaming remains poorly understood and this database will allow multiple researchers to leverage each other's work to perform novel and informative analyses. The resource developed here is potentially very important for the field. The manuscript and supplemental material describe some aspects of how the data is processed and the full details may be found in the database itself. The overall recommended minimal technical standards are consistent with general practice in the field. The database is well documented and the data are easily accessible. The authors also demonstrate how the dataset may be used by conducting analyses of power spectral density and automated sleep scoring. The results presented here are not particularly surprising from a scientific standpoint, but that is not the point. Rather, the authors successfully demonstrate that data from multiple studies can be analyzed through this database to answer questions about dreaming.

I do have two minor points to raise:

1) The authors carefully describe various types of sleep mentation and dreaming in the Background section of the manuscript. The behavioral data in the database is limited to dream report classifications as experience, experience without recall, and no experience. These coarse categories mean that much of the dream phenomenology described in the Background can not be deeply explored using the data in the database. It would have been ideal to have the text of the dream reports, but I understand that this may not be possible.

2) The authors have included MEG data in the database. While EEG and MEG are similar, they do measure slightly different

things. The authors should at least mention this fact briefly to soften the current implication (from phrases like “magneto/encephalography”) in the text that these methods are essentially interchangeable. It may be that MEG correlates of dreaming are different from EEG correlates of dreaming.

Overall, this project can advance our knowledge of sleep and I greatly look forward to seeing what results come from this work.

Reviewers comments in black

Our response in blue

Reviewer #1 (Remarks to the Author):

I confirm my positive opinion on the manuscript. The authors addressed my previous concerns.

We thank the reviewer for their positive appreciation.

Reviewer #2 (Remarks to the Author):

Apologies for the brief response:

1. I disagree this database will impact detection of anaesthetic awareness. I would remove all mention of anaesthesia. The pharmacology and interaction with surgical stimulation/artefact make it highly unlikely this database (of sleep studies) will help with that issue. That does not negate the utility of the database.

We have now removed the mention of anaesthetic awareness in the discussion as it is indeed far from the scope of brain dynamics that could be captured with this database.

2. This is underscored by the very low classification accuracy of the catch-22 results. I think the findings in this paper are largely incremental. Apologies I had not understood the previous submission did not require analyses, however I do not think the attached analyses, based on a limited number of electrodes, represent a major shift in the science of the study of consciousness.

We are glad that the scope of the paper now is better understood. Our intention was not to make a shift in consciousness research, but rather to provide a standardized database for dream and sleep studies. The analyses presented are merely demonstrative, and the potential of this database to decode dreams from EEG is very well exemplified by Moctezuma et al (<https://doi.org/10.1155/bmri/3585125>), where they achieved much better classification accuracy than our simple attempt using PSD and catch22.

3. That said, I don't have much else to critique and I think it is a worthy effort to make these data publicly available.

Thanks for your positive comments.

Reviewer #3 (Remarks to the Author):

The authors have constructed a large database of dream reports and EEG/MEG recordings by combining data from 20 studies. In addition, they propose to grow the database by incorporating new studies and offer a suggested set of technical details. This is a very exciting project for, as the authors note, this kind of data is difficult to collect at scale. Dreaming remains poorly understood and this database will allow multiple researchers to leverage each other's work to perform novel and informative analyses. The resource

developed here is potentially very important for the field. The manuscript and supplemental material describe some aspects of how the data is processed and the full details may be found in the database itself. The overall recommended minimal technical standards are consistent with general practice in the field. The database is well documented and the data are easily accessible. The authors also demonstrate how the dataset may be used by conducting analyses of power spectral density and automated sleep scoring. The results presented here are not particularly surprising from a scientific standpoint, but that is not the point. Rather, the authors successfully demonstrate that data from multiple studies can be analyzed through this database to answer questions about dreaming.

I do have two minor points to raise:

1) The authors carefully describe various types of sleep mentation and dreaming in the Background section of the manuscript. The behavioral data in the database is limited to dream report classifications as experience, experience without recall, and no experience. These coarse categories mean that much of the dream phenomenology described in the Background can not be deeply explored using the data in the database. It would have been ideal to have the text of the dream reports, but I understand that this may not be possible.

We agree with the reviewer's concern on the richness of the reports. While dream reports are not the norm on all the datasets, the datasets 'Oudiette_N1Data', 'Dream Database from Donders' and 'Kumral' have free reports that can be used for purposes more aligned with the various types of sleep mentation mentioned in the Background.

2) The authors have included MEG data in the database. While EEG and MEG are similar, they do measure slightly different things. The authors should at least mention this fact briefly to soften the current implication (from phrases like "magneto/encephalography") in the text that these methods are essentially interchangeable. It may be that MEG correlates of dreaming are different from EEG correlates of dreaming.

We appreciate the reviewer's comment and now we have mentioned that despite both techniques aiming to measure neural activity, the MEG and the EEG correlates of dreaming could be different due to the fundamental methodological differences. We have now included in the Background the following sentence:

"Also, different aspects of neural correlates of dreaming may be revealed by using MEG in addition to EEG because of their methodological differences."

Overall, this project can advance our knowledge of sleep and I greatly look forward to seeing what results come from this work.